# QUANTIZED BACK-PROPAGATION: TRAINING BINARIZED NEURAL NETWORKS WITH QUANTIZED GRADIENTS

**Itay Hubara, Elad Hoffer & Daniel Soudry**
Department of Electrical Engineering
Technion - Israel Institution of Technology
Haifa, Israel
`{itay.hubara,elad.hoffer,daniel.soudry}@gmail.com`

## ABSTRACT

Binarized Neural networks (BNNs) have been shown to be effective in improving network efficiency during the inference phase, after the network has been trained. However, BNNs only binarize the model parameters and activations during propagations. We show there is no inherent difficulty in training BNNs using "Quantized BackPropagation" (QBP), in which we also quantized the error gradients and in the extreme case ternarize them. To avoid significant degradation in test accuracy, we apply stochastic ternarization and increase the number of filter maps in a each convolution layer. Using QBP has the potential to significantly improve the execution efficiency (*e.g.*, reduce dynamic memory footprint and computational energy and speed up the training process, even after such an increase in network size.

## 1 INTRODUCTION

Deep Neural Networks (DNNs) achieved remarkable results in many fields making them the most common of-the-shelf approach for a wide variety of machine learning applications. However, using neural network (NN) algorithms on conventional general-purpose hardware, is highly inefficient. Much work has been done to reduce the size of networks. The conventional approach is to compress a trained (full precision) network. For example, Han et al. (2015) successfully pruned several state-of-the-art large scale network. More recent works demonstrated that by quantizing the parameters and intermediate activations during the training phase a more computationally efficient DNNs could be constructed (Gupta et al., 2015; Miyashita et al., 2016; Hubara et al., 2016a; Zhou et al., 2016).

This study proposes a more advanced learning technique, referred to as Quantized Back-Propagation (QBP), which for the backward pass we split the gradient streaming from the previous layer (i.e., error gradients) to quantized version that is served for the Back-Propagatin (BP) phase while still using the full precision for the update phase. In the extreme case QBP ternarizes the gradients and use only 1-bit for the weights and activations. In this case we refer to QBP networks as Ternarized Back-Propagation (TBP). Thus for the sequential backward phase TBP requires only Gated XNOR operations. Since the backward phase requires the same amount of multiplications compared to the forward pass, ternarizing the gradients is a crucial step toward faster training machines.

## 2 TERNARIZED BACK PROPAGATION

Although stochastic binarization is more appealing theoretically, since it ensure an unbiased estimator, previous work. Hubara et al. (2016a) demonstrated that it adds little to no gain in accuracy when binarizing the weights and activations. Therefore, in all our experiments we used the sign function for binarizing the weights and activations. Similar to previous work Hubara et al. (2016a); Mishra et al. (2017), we used the straight-through estimator (STE) approach that takes into account the saturation effect. The STE (which can be written as the derivative of a hard tanh) cancels gradients coming from neurons with absolute values higher than 1.

**Deterministic v.s. stochastic Ternarization**  In this study we have been experimenting with the following schemes: Deterministic ternarization:

$$x^t = \text{Tern}(x) = \begin{cases} sing(x) & |x| \geq th, \\ 0 & -th \leq x \leq th. \end{cases} \tag{1}$$

Stochastic ternarization:

$$x^t = \text{StcTern}(x) = \begin{cases} \text{sign}(x) & \text{w.p } \sigma_{th}(x), \\ 0 & \text{w.p } 1 - \sigma_{th}(x) \end{cases} \tag{2}$$

where:

$$\sigma_{th}(x) = \min(1, \frac{|x|}{2 \cdot th}) \tag{3}$$

In both case we first normalized each sample by its maximum absolute value which means that the normalization factor depend on the sample and the layer:

$$x_i^{norm} = \frac{x_i}{\max_i(|x_i|)} \tag{4}$$

and then applied the ternarization function Eq. 2 and multiplied each sample output after the MAC operation by its norm.

**Ternarizing only the error gradients**  In the back-propagation algorithm we recursively calculate the gradients of the loss function $\mathcal{L}$ with respect to $I_\ell$, the input of the $\ell$ neural layer

$$g_\ell = \frac{\partial \mathcal{L}}{\partial I_\ell} \tag{5}$$

starting from the last layer. Each layer needs to derive two sets of gradients to perform the recursive update. The layer activation gradients:

$$g_{\ell-1} = g_\ell W_\ell, \tag{6}$$

and the weights gradients

$$g_{W_\ell} = g_\ell^T I_\ell, \tag{7}$$

used to updated the weights in the $\ell$ layer. In this work we aim on ternarizing the gradients tensor $g_\ell$ only for the layer activation derivation (Eq. (6)) for the layer activation gradients derivation. Thus the gradients used for the weight gradients derivation are still in float. As previous works suggested Zhou et al. (2016); Hubara et al. (2016b) the use of full precision is unneeded and can be replace with 3-6 bits quantization. Ternarized Back-Propagation algorithm is detailed in Alg 1.

## 3  BENCHMARK RESULTS

In all our experiments we use Adam (Kingma & Ba, 2014) as our optimization method with mini-batch size of 256 and 32bit float precision batch-norm layers. We binarized the weights and activations for all layers except the last layer. Although Adam and Batch Normalization (which were kept in 32-bit precision) require some additional multiplications, we believe they can be avoided by using shift based algorithm as suggested by Hubara et al. (2016a).

The Multi-Layer-Perceptron (MLP) we trained on MNIST consists of 3 hidden layers with 2048 neuron each. With the following setting we achieved with TBP-BNN only 98.2% accuracy. We did not see any need to inflate the network as it was already very wide. This indicate a small degradation from BNN which received 1.4% accuracy. For CIFAR10 dataset we used a VGG-like network similar to the one suggested by Hubara et al. (2016a) with the same hyper-parameters used in the original work. The results demonstrate that if we inflate the convolutional filters by factor of 3 we can achieve similar results as the BNN and full precision models achieved 9.53% accuracy. This is in accordance with previous finding (Mishra et al., 2017), that found that widening the network can mitigate accuracy drop inflicted by low precision training. To make sure this is not a unique case for BNN we also applied TBP on ResNet-18 in which we recived 10.8% accuracy after inflating it by factor of 5.

For ImageNet dataset, we used on AlexNet model inflated by factor of 3. Similar to previous work and to ease the comparison we kept first and last layer in full precision. With 4-bit activation and 4-bit gradients QBP converged to 53.3% top-1 accuracy and 75.84% top-5 accuracy. By using only 2bit activation QBP reached 49.6% top-1 accuracy and 73.1% top-5 accuracy. We are currently working on more advanced typologies such as ResNet-50 model He et al. (2016).

### 3.1 DISCUSSION AND ADDITIONAL EXPERIMENTS

To shed light on what TBP works and what is yet to be solved we conducted additional set of experiments on the CIFAR-10 dataset with both ResNet-18 and the BNN like typologies.

**Ternaizring both phases.** Ternarizing both stages results with completely MAC free training. However, our results show that without enabling at least 3bit precision for the update stage the model reaches only approximately 80% accuracy. Indicating that the ternarization noise is to high thus distorts the update gradients direction. If we stop the gradients ternarization once the accuracy ceases to increase, the convergence continues and the accuracy increases to the same accuracy as TBP. Thus, TBP can also be used to accelerate training of BNN networks by first training it with TBP and then, for the last couple of epochs, continue training with full precision gradients.

**Multiple stochastic ternarization sampling.** To alleviate the need for float MAC operation in the update phase we suggest to run gated XNOR operation multiple times each with different stochastic sample the ternarized tensor and average the results. As expected the accuracy improves with the amount of sampling. To find the number of samples needed for each layer we adopted a similar geometrical approach as suggested by Anderson & Berg (2017) and measured the correlation coefficient ($R$) between the update gradients received with and without ternarization. Our experiments indicate that more samples are required for the first two convolution layer (12 samples) while the rest of the layers need approximately 6 samples. Using this configuration keeps the correlation coefficient above 0.7 and results with 87.5% accuracy.

**Ternarizing only the gradients used for the update phase.** Ternarizing only the gradients used for the update phase while keeping the back-propagation gradients in float converged well and received 86.3%. However, ternarizing the update phase is much less effective in accelerating DNNs. This result is not surprising as Wen et al. (2017) showed that for distributed training the gradients update can be ternarized to reduce communication overhead.

**One normalization factor for the entire gradient tensor.** Using only one normalization factor for the entire gradient tensor recieved almost the same accuracy as classic TBP and, 86.5% on CIFAR-10 using ResNet-18. Since we use ternarization and not binarization the zeros helps preserving each sample importance hence preserve the update gradient direction.

**Deterministic ternarization.** Using deterministic ternariztion simply does not work. In our experiments the update accuracy cease to improve after the first couple of update stages and achieves only approximately 20% accuracy. This experiment indicates that the noise added by the gradients quantization process must have a bias equal to zero

## 4 DISCUSSION AND CONCLUSION

In this study we introduce Quantized Back-Propagation (QBP). This method requires sufficiently wide networks to work well. However, Mishra et al. (2017) suggested that in practice, the effectiveness of reduced precision neural networks highly depends on the hardware that runs the low precision operation. In Mishra et al. (2017), Section 5.1, it was demonstrated, on a dedicated ASIC, that BNNs can achieve an improvement of three orders of magnitude (figures 3f and 3g in Mishra et al. (2017)) in compute and energy performance. This suggests that even if we increase the number of operations by modestly inflating the network (as we do here) we can still obtain a much more efficient design. Thus, QBP (and by extension TBP) still posses the potential to drastically reduce memory size and accesses, and substantially increase the power efficiency.

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
