# OpenReview forum: "Quantized Back-Propagation: Training Binarized Neural Networks with Quantized Gradients"
_ICLR.cc/2018/Workshop — Reject_

### Official Review · AnonReviewer2 · 2018-03-08
**good work but not well written**

**Rating:** 5
**Confidence:** 4

**Review:**

The authors explored the possibility of using reduced precision in backpropagation. Experimental results support the idea of QBP or even TBP, showing that models trained with quantized gradients are able to get good performances.

The manuscript needs much more work to be in a good form. For example,
0. The term MAC has appeared before its definition.
1. On page 2, end of Section 2, the authors mentioned Algorithm 1, but there is no algorithm table in the paper.
2. "for the layer activation gradients derivation" appeared twice in the same sentence.

Also, there is already a related work which is exactly doing quantized back propagation: Lin, Z., Courbariaux, M., Memisevic, R. and Bengio, Y., 2015. Neural networks with few multiplications. arXiv preprint arXiv:1510.03009. It would be great if the authors could describe the advantages of the current method over the previous one.

---

### Comment · AnonReviewer1 · 2018-03-08
**good paper**

This paper tackles quantization of the backward-pass in DNNs. The paper focuses mostly on vision/CNN workloads.

The idea is interesting - most works till now have looked into very low-precision for the forward pass. For backward-pass, FP16 gradients are SOTA now. This paper looks at precision below 16bits - ternary and 4bits.

Writing quality of the paper can be significantly improved. For example, sin'g' in equation-1, no algorithm description (Alg 1.) in end of Section 2, etc.

Also, the paper mentions backward pass requires the same amount of multiplications as the forward pass - this is incorrect, backward pass requires 2x more compute than forward pass (one for weight in current layer and one for loss propagation to the previous layer).

I can buy into the argument of training at low-precision (forward+backward pass) for most of the epochs and then train with full-precision the last few epochs, i.e. accelerated training time. This paper sets this direction.

Also, it would be good to get more data-points on why stochastic ternarization helps - why noisy gradients help and why accuracy does not go up without this.

Overall, the first steps in this paper and initial results are promising.

---

### Decision · Program_Chairs · 2018-03-20
**ICLR 2018 Workshop Acceptance Decision**

**Decision:**

Reject

**Comment:**

Based on the reviews, this paper has not been accepted for presentation at the ICLR workshop. However, the conversation and updates can continue to appear here on OpenReview.